# Multiscale segmentation using hierarchical phase-contrast tomography and deep learning

Yang Zhou[1], Shahab Aslani[2,3], Yousef Javanmardi[1], Joseph Brunet[1,4],
David Stansby[1,5], Saskia Carroll[1], Alexandre Bellier[6], Maximilian Ackermann[7,8,9],
Paul Tafforeau[4], Peter D. Lee[1], Claire L. Walsh[1]*

**1** Multiscale X-ray Imaging (MXI) Lab, Department of Mechanical Engineering, University College London, London, United Kingdom, **2** Satsuma Lab, Hawkes Institute, University College London, London, United Kingdom, **3** Department of Respiratory Medicine, University College London, London, United Kingdom, **4** European Synchrotron Radiation Facility, Grenoble, France, **5** Advanced Research Computing Centre, University College London, London, United Kingdom, **6** Univ. Grenoble Alpes, Department of Anatomy (LADAF), AGEIS, CIC INSERM, Grenoble, France, **7** Institute of Anatomy, University Medical Center of the Johannes Gutenberg University Mainz, Mainz, Germany, **8** Institute of Pathology, Uniklinik RWTH Aachen, Aachen, Germany, **9** Institute of Pathology and Department of Molecular Pathology, Helios University Clinic Wuppertal, Wuppertal, Germany

* c.walsh.11@ucl.ac.uk

## Abstract

Biomedical systems span multiple spatial scales, encompassing tiny functional units to entire organs. Interpreting these systems through image segmentation requires the effective propagation and integration of information across different scales. However, most existing segmentation methods are optimised for single-scale imaging modalities, limiting their ability to capture and analyse small functional units throughout complete human organs. To facilitate multiscale biomedical image segmentation, we utilised Hierarchical Phase-Contrast Tomography (HiP-CT), an advanced imaging modality that can generate 3D multiscale datasets from high-resolution volumes of interest (VOIs) at ca. 1 $\mu m$/voxel to whole-organ scans at ca. 20 $\mu m$/voxel. Building on these hierarchical multiscale datasets, we developed a deep learning-based segmentation pipeline that is initially trained on manually annotated high-resolution HiP-CT data and then extended to lower-resolution whole-organ scans using pseudo-labels generated from high-resolution predictions and multiscale image registration. As a case study, we focused on glomeruli in human kidneys, benchmarking four 3D deep learning models for biomedical image segmentation on a manually annotated high-resolution dataset extracted from VOIs, at 2.58 to ca. 5 $\mu m$/voxel, of four human kidneys. Among them, nnUNet demonstrated the best performance, achieving an average test Dice score of 0.906, and was subsequently used as the baseline model for multiscale segmentation in the pipeline. Applying this pipeline to two low-resolution full-organ data at ca. 25 $\mu m$/voxel, the model identified 1,019,890 and 231,179 glomeruli in a 62-year-old donor without kidney diseases and a 94-year-old hypertensive donor, enabling comprehensive morphological analyses, including cortical spatial statistics and glomerular distributions, which aligned well with previous

**Data availability statement:** All code is publicly available at https://github.com/UCL-MXI-Bio/2025-zhou-hipct-hierarchical-segmentation, and high-resolution annotated data can be accessed at https://doi.org/10.5281/zenodo.15397768.

**Funding:** This work was supported by the Chan Zuckerberg Initiative DAF, an advised fund of Silicon Valley Community Foundation (DAF2022-316777 and CZIF2024-009938 to YZ, YJ, JB, DS, SC, CLW), NIH BRAIN Initiative CONNECTS program via National Institute of Neurological Disorders and Stroke (NINDS) and National Institute of Mental Health (NIMH) (UM1-NS132358 to YZ, DS, CLW), Wellcome Trust (310796/Z/24/Z to YZ, DS, CLW), International Alliance for Cancer Early Detection, an alliance between Cancer Research UK, Canary Center at Stanford University, the University of Cambridge, OHSU Knight Cancer Institute, University College London and the University of Manchester (EDDAPA-2023/100002 to SA), Royal Academy of Engineering (RAEng) (CiET1819/10 to PDL), and CIFAR MacMillan Multiscale Human Fellowship. The funders had no role in study design, data collection and analysis, decision to publish, or preparation of the manuscript.

**Competing interests:** The authors have declared that no competing interests exist.

anatomical studies. Our results highlight the effectiveness of the proposed pipeline for segmenting small functional units in multiscale bioimaging datasets and suggest its broader applicability to other organ systems.

## Author summary

Understanding how the biosystems work requires studies on not only entire organs but also small functional structures within them. It is challenging to analyse these tiny structures across a complete organ at different scales, because most medical imaging techniques only focus on a single scale of detail at a time. In this study, we used a synchrotron-based X-ray imaging technique called Hierarchical Phase-Contrast Tomography, which can scan human organs at multiple levels of detail from regions of interest at ca. 2 $\mu m$/voxel to full organs at ca. 25 $\mu m$/voxel. Based on the hierarchical dataset, we developed a deep learning-based method that is trained to recognise small structures in the high-resolution images and then applies that knowledge to identify the same structures in lower-resolution scans of the entire organ. Using the human kidney as an example, we demonstrated that our method can accurately detect glomeruli, tiny units responsible for filtration, throughout the whole human kidney. This approach opens up new possibilities for studying the structure and distribution of functional units in 3D across complete organs, which would improve our understanding of health, disease progression, and organ function.

## Introduction

Biological systems are organised across multiple 3D scales from macro-scale complete organs to micro-scale functional units such as alveoli and nephrons. These units are considered the smallest structures of organs that perform a unique physiological function. Understanding them across entire macro-scale organs enables analysis of spatial functionalities, distributions and variations among healthy human and pathological contexts [1]. However, such multiscale studies are limited by the lack of 3D image datasets that simultaneously provide both high-resolution micro-scale regions and macro-scale whole organ coverages [2]. Traditionally, micro-level features captured by 2D tissue sections [3] do not present 3D structure, while clinical 3D imaging techniques are limited to macro-level structures due to their resolution constraints [4]. x Recent development of Hierarchical Phase-Contrast Tomography (HiP-CT) [5], a synchrotron-based X-ray technique, images intact human organs at ca. 20 $\mu m$/voxel down to micro-scale volume-of-interests (VOIs) at ca. 1 $\mu m$/voxel. This hierarchical property facilitates multiscale biomedical image research. Generally, due to the limitations of longer scanning, high X-ray radiation dose, and large data volumes, scanning the complete organ at the highest possible resolution (e.g., 1 $\mu m$/voxel) is not routinely desirable and efficient. HiP-CT instead typically provides high resolution in smaller VOIs and lower resolution overviews of whole organs. These hierarchical 3D VOIs allow visualising and aligning tiny functional units across different scales.

Therefore, by utilising HiP-CT data, we aimed to develop a deep learning-based segmentation pipeline for multiscale biomedical images, which leverages the resolution hierarchy of HiP-CT and deep learning to enable the analysis of micro-scale functional units across macro-scale complete organs. Previous studies in [6–8] achieved 3D vasculature segmentation across multiple scales of human kidneys using manually annotated HiP-CT datasets. However, manually annotating HiP-CT data at different scales is time-consuming and labour-intensive, specifically for small functional units that can only be observed at high-resolution data. To surmount this problem, we only labelled several VOIs from high-resolution data and trained a segmentation model on this limited high-resolution dataset. Then, we proposed a multiscale segmentation pipeline that performs multiscale registration and pseudo-labelling to hierarchically propagate the segmentation capability of small functional units to lower-resolution VOIs, ultimately to the complete organ scans. The multiscale registration aims to align the HiP-CT volumes across scales, from which the transformation can be applied to high-resolution predictions as pseudo-labels for lower-resolution data to fine-tune the model in the previous resolution. Since the pseudo-labels are derived from higher-level predictions, we also proposed a post-processing strategy to remove the false positives.

Here, to evaluate the proposed deep learning-based multiscale segmentation pipeline on HiP-CT, we used glomeruli segmentation as a case study. The glomerulus is one of the small functional units in the kidneys. It is approximately spherical-shaped capillary network of ca. 200 $\mu m$ in diameter, situated within the cortex of the human kidney, and plays a key role in blood filtration and autoregulation [9]. Although the glomeruli are difficult to unambiguously manually annotate on complete kidney HiP-CT scans at lower resolutions (e.g., 20 $\mu m$/voxel), they are clearly presented in the high-resolution HiP-CT scans, as shown in Fig 1A, single healthy kidney contains an estimated 330,000 to 1,400,000 glomeruli [10]. Prior glomeruli segmentation has primarily relied on 2D high-resolution (e.g., 0.85 $\mu m$/pixel) histological images, which limit 3D morphological analysis due to distortions and incomplete volumetric information [11]. In this study, our proposed multiscale segmentation pipeline, together with HiP-CT, enables comprehensive 3D segmentation and morphological analysis of glomeruli across complete human kidneys.

In summary, the key contributions of this study are: (1) We present a 3D HiP-CT multiscale glomeruli segmentation dataset, including manual annotation on high-resolution VOIs from four intact human kidneys; (2) We propose a multiscale segmentation pipeline that can propagate the segmentation capability from annotated high-resolution HiP-CT VOIs to low-resolution intact organs volumes through multiscale registration and fine-tuning with pseudo-labels; (3) We benchmark four state-of-the-art 3D deep learning models: VNet [12], UNETR [13], SwinUNETR [14] and nnUNet [15] on high-resolution annotated data. The best-performing model is selected for correlative segmentation across scales; (4) We apply the full pipeline to two complete human kidneys from a 62-year-old donor without kidney diseases and a 94-year-old hypertensive donor, enabling downstream 3D morphological analysis of segmented glomeruli. The results are aligned with current anatomical studies.

## Related works

Biomedical image segmentation is important to generate pixel-wise (2D) and voxel-wise (3D) labels of distinct structures, potentially aiding clinical applications such as automated diagnosis [16,17]. Previous manual and semi-automated segmentation techniques are time-consuming and labour-intensive. With the advancement of deep neural networks, early automated methods aimed to solve 2D segmentation problems. Fully connected networks (FCN) [18] were the first end-to-end networks that could generate direct and dense segmented predictions at an arbitrary input size. However, better FCN performance is limited by the depth of the network due to the gradient vanishing problem [19]. To alleviate this, short skip connections [20], similar to residual connections, were introduced. FCNs also have difficulty preserving fine-grained details during upsampling without effective network structures. Therefore, [21] proposed a symmetric encoder-decoder structure, the U-Net, to propagate the context information in the upsampling stage. UNet++ [22] replaced the U-Net direct skip connections with nested and dense skip connections to enrich the feature maps from the encoder network to improve the segmentation performance.

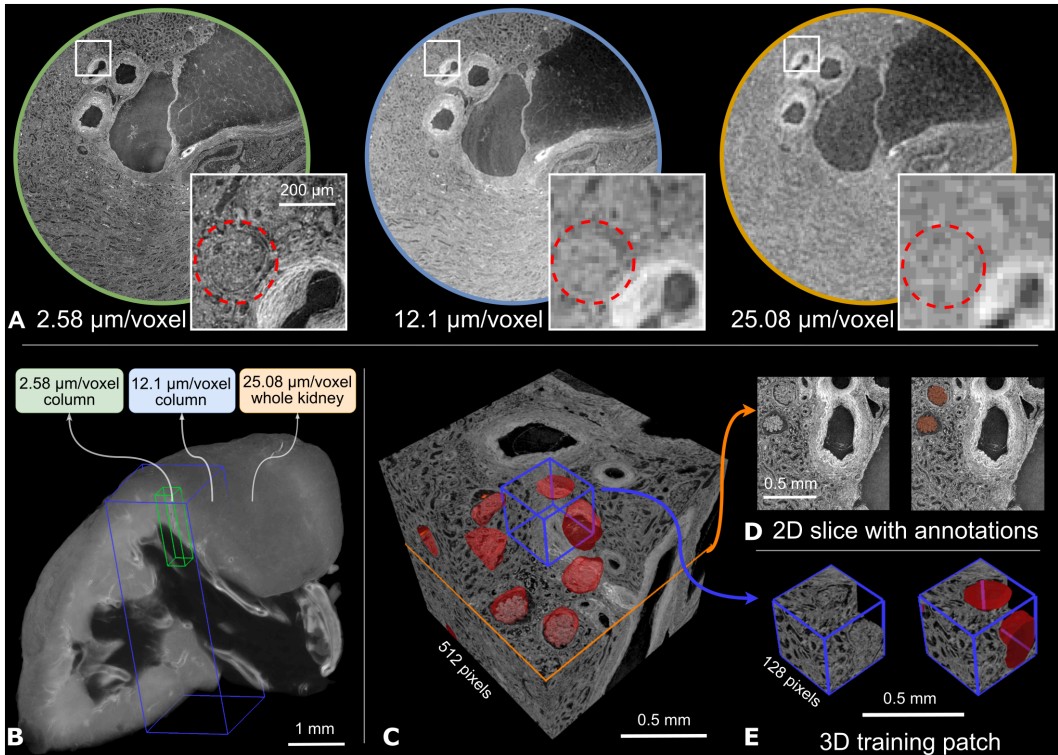

**Fig 1. Multiscale HiP-CT dataset for glomeruli segmentation from the sample of LADAF-2020-17 left kidney.** A: Visualisation of 2D slices from the same region at three different resolutions: 2.58 $\mu m$/voxel [32], 12.1 $\mu m$/voxel [33], and 25.08 $\mu m$/voxel [34]. Inset squares highlight the same glomerular region across resolutions. B: 3D rendering of the full kidney at 25.08 $\mu m$/voxel, with cuboids indicating the locations of two selected higher-resolution volumes at 12.1 $\mu m$/voxel and 2.58 $\mu m$/voxel. C: Manually annotated training volumes in dimension of $512^3$, where glomeruli are marked in red. D: Example 2D slice of the training cube with glomeruli annotated in red. E: Cropped 3D Training patches in dimension of $128^3$, used as inputs for neural network training.

Considering biomedical images are typically volumetric, and training a model on 2D slices only does not allow for the learning of 3D morphological features, V-Net [12] refined the 2D U-Net with 3D convolutional kernels and added residual connections in downsampling to accelerate convergence on 3D data. Since biomedical segmentation benefits from wide contextual information, non-local U-Net [23] used an aggregation block based on a non-local attention mechanism [24] to involve the global information without a deep encoder network. However, it is difficult to transfer a universal network structure across 3D biomedical images with different properties, like voxel spacing, and different modalities. Those properties illustrate the physical distances (spacing) and statistical differences (modalities) for the input images [25]. Thus, [15] proposed nnUNet, an automated biomedical image segmentation pipeline, involving pre-processing, network structure configurations, and post-processing for any incoming dataset. To improve small object segmentation, omitted by U-shape network structures, KiU-Net [26] transferred the input to a higher spatial dimension and downsampled it to obtain the same-size output. Despite their success, these networks fail to capture long-range spatial dependencies and global context in challenging datasets.

Inspired by language transformers that can preserve long-range information, Vision Transformers (ViT) [27] allow training a Transformer network with image patches as tokens, achieving promising performances on classification tasks. TransUNet [28] firstly incorporated the Transformer self-attention block into CNN as a hybrid CNN-Transformer network, showing the potential on the 2D biomedical image segmentation task. For 3D image segmentation, UNETR, developed by [13],

directly applied the Transformer as an encoder and used 3D patches as inputs. SwinUNETR [29] integrates the hierarchical encoder from Swin Transformer to achieve better segmentation performances.

Although the above networks are promising for HiP-CT segmentation, pseudo-label propagation between different resolutions requires fine-tuning techniques. A previous study in [30] showed that re-training the network on different scales transferred the classification performance of skin lesions across different resolutions. MorphHR [31] fine-tuned the network with morphed layers and successfully applied the model that is trained on natural images to high-resolution mammogram images for classification tasks. However, fine-tuning a network for segmentation tasks and multiscale biomedical images with different voxel sizes remains a research gap.

## Data

HiP-CT generates multiscale 3D images of intact human organs from ca. 20 $\mu m$/voxel to zoomed VOIs at ca. 1 $\mu m$/voxel. This hierarchical feature enables morphological studies on small functional units, e.g., glomeruli in the diameter size range of 100 $\mu m$ to 200 $\mu m$, and statistical analysis across whole organs. As HiP-CT creates nested VOIs of increasing resolution without the need to section the organ physically, alignment between different resolutions is relatively straightforward. Fig 1A shows registered slices in different resolutions of one kidney, and Fig 1B depicts the 3D visualisations of registered VOIs in this kidney.

The multiscale HiP-CT glomeruli dataset involves three different resolutions: high resolutions of ca. 2.58 to 5.2 $\mu m$/voxel, intermediate resolution of ca. 12 $\mu m$/voxel, and low resolution of ca. 25 $\mu m$/voxel. To reduce the data size and computational resource needs, the high-resolution and intermediate-resolution data were binned by a ratio of two. Four different kidneys from three donors were used, as shown in Table 1, prepared in ethanol before being scanned at the European Synchrotron Radiation Facility (ESRF) with HiP-CT (see Methods for details). Manual annotation was conducted on 40 cubes with dimensions of $512^3$ from high-resolution volume-of-interests (VOIs) of those four kidneys, varying from 2.58 $\mu m$/voxel to 5.2 $\mu m$/voxel (binned by 2 to reduce the data size). To annotate the glomeruli on the high-resolution VOIs, the annotators evaluated several adjacent slices to label them according to obvious 3D morphological features.

Dataset statistics are shown in Table 1. Due to computational resource limitations, the manually annotated cubes of $512^3$ were cropped into non-overlapping 3D patches of $128^3$ voxels for training. Fig 1C shows one of the labelled cubes in both 2D slices and 3D patches. We aimed to follow a ratio of 9:1 to split the train-test dataset based on the original cubes of dimensions $512^3$. Considering the imbalance in the number of cubes annotated for each kidney, at least one cube was selected for the test dataset from every kidney. The statistics for data splits are shown in Table 1.

## Methods

### Ethics statement

Four human kidneys used in this study were obtained with ethical approvals from three donors, LADAF-2020-27, LADAF-2021-17, and S-20-28. LADAF-2021-17 left kidney, LADAF-2021-17 right kidney, and LADAF-2020-27 left kidney were collected from the donors who had provided written consent for body donation to the Laboratoire d'Anatomie des Alpes

**Table 1**. Information and statistics of HiP-CT glomeruli segmentation dataset for training based on 9:1 train-test split across 4 samples.

| Samples | Information | Training | | Testing | |
|---|---|---|---|---|---|
| | | Cubes | Patches | Cubes | Patches |
| 5 $\mu m$ S-20-28 [35–41] | Male, aged 84, prepared with ethanol | 18 | 831 | 2 | 119 |
| 5.2 $\mu m$ LADAF-2021-17 Left Kidney [42] | Male, aged 63, prepared with ethanol | 6 | 187 | 1 | 10 |
| 5.2 $\mu m$ LADAF-2021-17 Right Kidney [43] | Male, aged 63, prepared with ethanol | 8 | 226 | 1 | 13 |
| 2.58 $\mu m$ LADAF-2020-27 Left Kidney [32] | Female, aged 94, prepared with ethanol | 3 | 86 | 1 | 26 |
| Total | - | 35 | 1330 | 5 | 168 |

Françaises (LADAF) before death. The ethics committee of these specimens was the Comité d'éthique, scientifique et pédagogique du Département d'Anatomie de l'Université Grenoble Alpes (Ethics, Scientific and Educational Committee of the Department of Anatomy at the University Grenoble Alpes). LADAF is an officially authorised body donation centre under the supervision of the French Ministry of Higher Education and Research. The kidney of S-20-28 was obtained after a clinical autopsy at the Hannover Institute of Pathology at Medizinische Hochschule, Hannover. The ethics committee was the Medical School of Hannover, Germany, under approval number 9621 BO K 2021. The transport ethics approval was obtained from the French Ministry of Health. All ethics statements were consented in writing.

## Data acquisition

HiP-CT is an X-ray phase-contrast propagation technique using the Extremely Brilliant Source (EBS) from the European Synchrotron Radiation Facility (ESRF). Prior work [5,44] describes the sample preparation and imaging procedure. All the organs were transported to ESRF for scanning. HiP-CT scan parameters and relevant tomographic information are presented in the Table A in S1 Text.

## Multiscale segmentation pipeline

As shown in Fig 2, the proposed multiscale segmentation pipeline employs hierarchical cycles to propagate the segmentation from high-resolution data, where the glomeruli are clearly visible, to low-resolution complete kidneys. Each hierarchical cycle began with training data preparation (Fig 2A) and proceeded through five steps: data pre-processing (Fig 2B), training/fine-tuning the deep neural network (Fig 2C), prediction post-processing (Fig 2D), multiscale registration (Fig 2F), and pseudo-labelling lower-resolution data (Fig 2G). In the first cycle, we benchmarked four state-of-the-art 3D segmentation models on manually annotated high-resolution data and selected the best-performing model for all subsequent steps. Because manual annotation at lower resolutions is challenging, we introduced a pseudo-labelling strategy that leverages high-quality predictions from the previous resolution cycle as pseudo-labels, transferred through multiscale registration and prediction post-processing. The following sections describe each step of this pipeline in detail.

**Data pre-processing.** As the voxel intensity of HiP-CT data is non-quantitative and can vary substantially between samples due to imaging setup, anatomical variation, sample preparation, etc., it is essential to normalise datasets across different samples. To do this, we applied contrast-limited adaptive histogram equalisation (CLAHE) [45] to each sample, which decreases the noise amplified in the near-contrast regions (see Fig A in S1 Text).

The original HiP-CT images form large datasets (typically 250 GB - 1 TB per volume). Training the deep neural networks on such big datasets is challenging, and following preliminary experimentation [6,7], we found that conversion of the data from 16-bit to 8-bit after application of CLAHE reduced the memory requirements and allowed for better optimisation of the pipeline, leading to better overall segmentation outputs. After that, the division of the original labelled cubes ($512^3$ voxels) into non-overlapping cubes of $128^3$ was performed. This size enabled training a model with a larger batch size while being able to keep a whole glomerulus in the volume of interest. To keep the consistency and the network structure in the fine-tuning, the same preprocessing techniques were applied to the lower-resolution data in the pipeline.

**Training with manual labels.** Since HiP-CT generates 3D isotropic data, we tested four 3D segmentation deep learning models: VNet, UNETR, SwinUNETR, and nnUNet in the first hierarchical cycle of high-resolution data with manual labels. The performance of these four models was evaluated by the average Dice score, a metric measuring the overlap area between predictions and ground truths, using 5-fold cross-validation. In this cycle, the highest performing model was selected as the baseline model to then perform the correlative segmentation at subsequent hierarchical cycles with lower-resolution datasets. All the models were developed with PyTorch [46] 2.0.1, and all the experiments were performed on four Tesla V100 GPUs, and the inference was on one NVIDIA RTX A6000 GPU.

**Post-processing.** In this pipeline, post-processing was crucial for eliminating false positives from predicted glomeruli before registration and propagation as pseudo-labels to lower-resolution data. It improved the quality of the automated

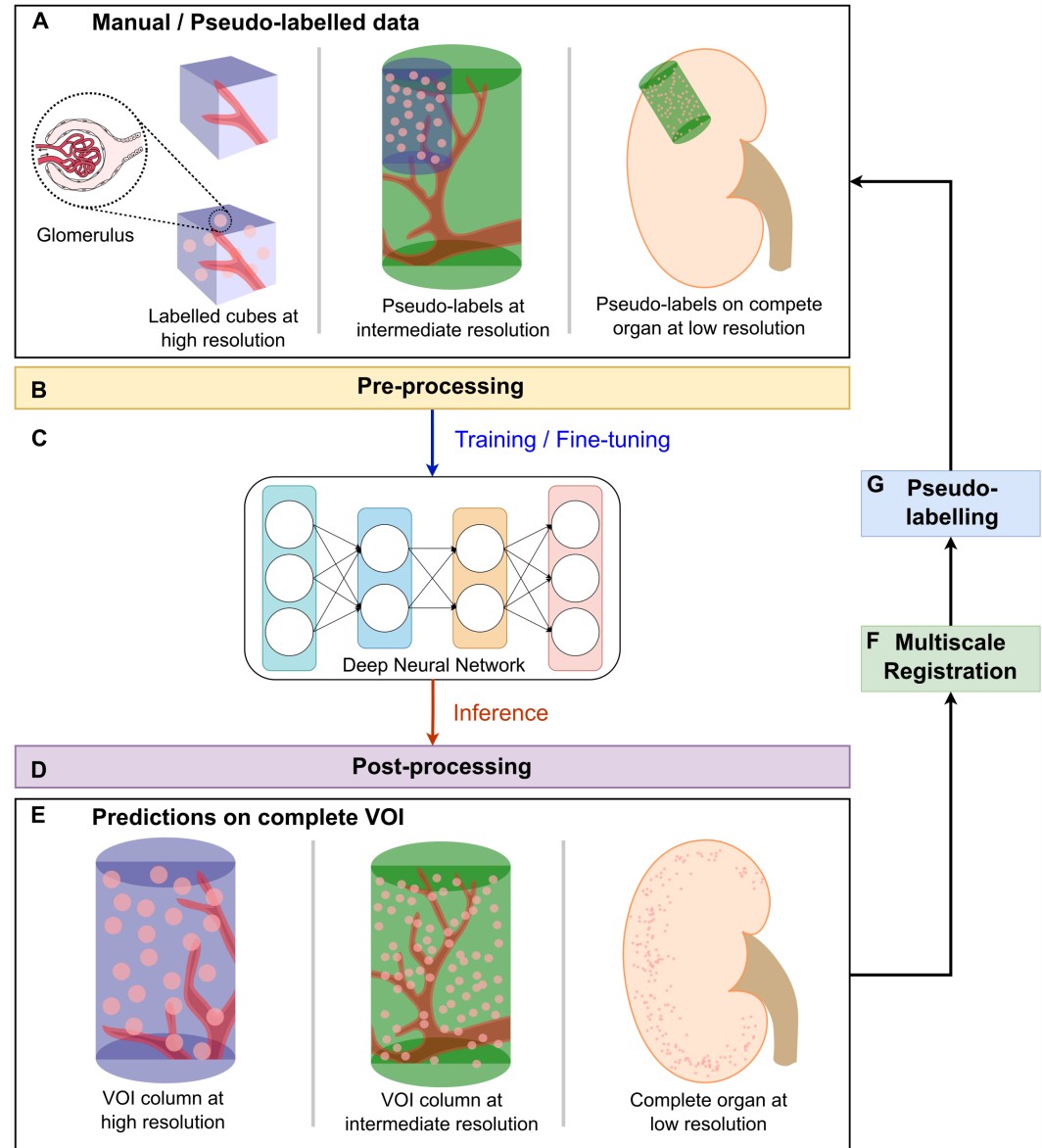

**Fig 2**. **Overview of the proposed multiscale segmentation pipeline using glomeruli as a case study.** This pipeline enables segmentation of tiny functional units across multiple resolutions of HiP-CT data. A: Input data at multiple HiP-CT resolutions, with corresponding manual annotations or pseudo-labels; B: Pre-processing steps, including contrast enhancement and patch extraction, to improve image quality and address memory constraints; C: Training or fine-tuning of deep neural networks using the labelled HiP-CT data; D: Post-processing to eliminate false positives based on intensity and morphological criteria; E: Final post-processed predictions on the complete HiP-CT volume of interest (VOI); F: Multiscale registration aligning higher-resolution VOIs with lower-resolution scans; G: Generation of pseudo-labels for low-resolution data by propagating high-resolution predictions through the registration transforms.

labels used in fine-tuning the network. In each hierarchical cycle, after generating the glomeruli predictions for the complete VOI from which the training cubes had been extracted. A large number of false positives can be found in the inferred predictions. To remove these false positives, we generated instance segmentations where each predicted glomerulus has associated intensity and morphological properties. We then selected several representative parameters, such as intensity variance, roundness, and size, and generated histograms of these properties to filter the outliers.

Considering the organ shrinkage due to dehydration during preparation, the minimum radius of a glomerulus is set to 62 $\mu m$ (volume= $1 \times 10^6 \mu m^3$), which is close to the smallest measurement from MRI in [47] of 72 $\mu m$ (volume= $1.6 \times 10^6 \mu m^3$). Apart from this fixed minimum size, other thresholds for the selected parameters are automatically set based on the percentiles of the aggregate data for all glomeruli. From these threshold bounds, Latin Hypercube Sampling (LHS) [48] was then used to explore the parameter space to optimise the combination of post-processing. 20 parameter sets were generated through LHS, and each parameter set was evaluated using Dice scores on all labelled cubes and an additional empty cube containing only fat, where no glomeruli should be present, and where the majority of the false positives were found. The properties and the threshold criteria are listed in Table 2 in the Results Section.

**Multiscale registration.** Intact human organs at lower resolution are scanned first, followed by the high-resolution VOIs. Therefore, the multi-resolution VOIs must be registered to the complete organ volume to align the features during correlative model fine-tuning using higher-resolution predictions as pseudo-labels.

The registration was implemented utilising SimpleITK [49] and Mattes Mutual Information [50,51] as a similarity metric over optimisation, between the lower-resolution fixed volume and the higher-resolution moving volume. The process started with two manually selected common points in each solution as a fixed centre. The common points are identical features among multi-resolution data recognised by experienced experts. With the fixed centre, the registration process started with finding the $z$-rotation angle between the high-resolution and the low-resolution volumes. The process involves two consecutive exhaustive searches - 360 degrees with 2 degrees as a step and 5 degrees with 0.1 degrees as a step. By searching on a large range and then a smaller range, the registration process can quickly optimise to find the z-rotation angles. After that, the $xy$-plane rotation angle and scaling factor are optimised by the Limited-memory Broyden–Fletcher–Goldfarb–Shanno (LBFGSB) algorithm with 2000 iterations. The registration results are presented in the Fig B in S1 Text.

**Pseudo-labelling and fine-tuning.** We then introduced pseudo-labelling to automatically generate training pseudo-labels for lower-resolution data that are difficult to manually annotate. The pseudo-labels were the predictions of a higher-resolution data volume from the previous hierarchical cycle. Those predictions were registered to the lower-resolution data volume that was cropped into cubes with a size of $128^3$ as a training dataset, consistent with the cube size for higher resolution data. Then, the trained model was fine-tuned on the new data, adapting it to lower-resolution data. The hyperparameters of the model used for fine-tuning are the same as those used for the high-resolution training stage.

For the lowest-resolution data, ca. 25 $\mu m$/voxel, the glomeruli features are subtle in lower-resolution data due to the high voxel size (see Fig C in S1 Text). To obtain the best segmentation performance with the pseudo-labels, we explored different training strategies, such as (1) training from scratch vs fine-tuning, (2) changes in learning rates, which are presented in the Fig D in S1 Text.

**Deep neural network optimisation.** The deep neural networks were optimised by an ensemble cost function $L$ of a cross-entropy loss $L_{CE}$ and a weighted Dice loss $L_{Dice}$ for glomeruli segmentation as shown in Eq 1.

$$L = L_{CE} + \beta L_{Dice}$$
$$= (1 - \sum p(x) \log q(x)) + \beta(1 - \frac{2 \sum p(x) \sum q(x) + \varepsilon_1}{\sum p(x) + \sum q(x) + \varepsilon_2}),$$

(1)

where $p(x)$ and $q(x)$ are the prediction and ground truth at a given voxel coordinate $x$, and the sums are over all voxels in a patch. $\varepsilon_1$ and $\varepsilon_2$ are small constant values, set to $1 \times 10^{-5}$ in this work, to avoid division by zero in the Dice loss. $\beta$ is the weight of the Dice loss relative to the cross-entropy loss. To balance the loss contributions from regional overlap and pixel-level accuracy, we set $\beta$ as 1 in all the experiments.

Cross-entropy loss is widely used in segmentation tasks to quickly train the network based on pixel-wise error and stable gradients. However, it tends to place larger weights on the larger objects while neglecting small ones [52]. Considering that the glomeruli segmentation consists of two classes (glomeruli or non-glomeruli) with the glomeruli class appearing sparsely, we added the additional Dice loss with a weight $\beta$ to the overall loss. The Dice loss measures the overlap areas

between predictions and labels, and is inherently designed for imbalanced classes. Therefore, our cost function enables the networks to learn voxel-wise and regional overlap information.

## Results

### Glomeruli segmentation on high-resolution labelled data

The performance of the four initial network architectures was tested on the highest resolution data and compared using Dice scores across 5-fold cross-validation through training, validation, and test datasets. Fig 3A shows the average Dice scores and their standard deviations revealed by the error bar after training each model with 150 epochs. After that, the Dice scores of the test dataset were calculated from the best models of each method.

As shown in Fig 3A, during training, nnUNet had the highest average Dice score of 0.923, followed by SwinUNETR of 0.837, UNETR of 0.809, and VNet of 0.770. However, VNet achieved a higher Dice score than the Transformer models in validation. As for the test dataset, we performed sliding windows inference using the best models of each method on the complete annotated data in the size of $512^3$, without cropping them to the size of training and validation patches of $128^3$. This improved the segmentation performance compared to the Dice scores of the validation data.

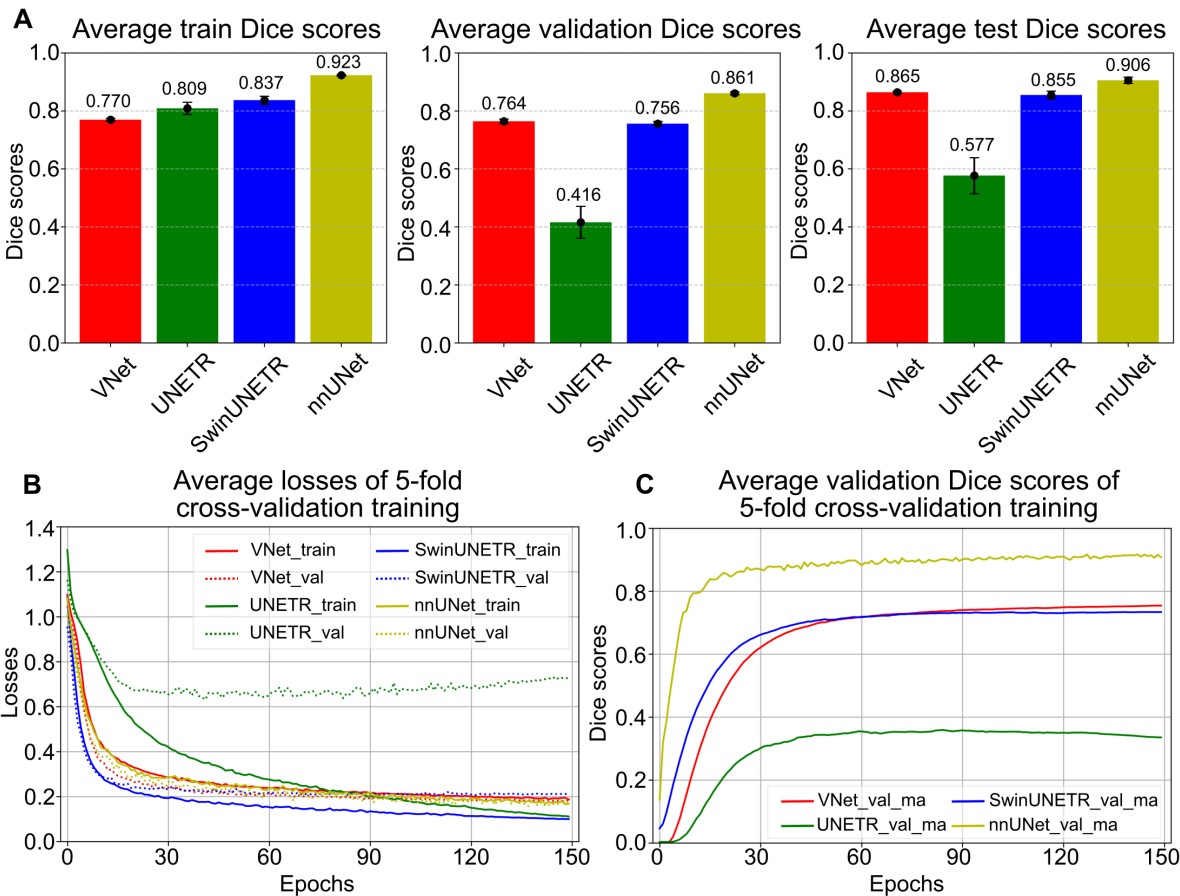

**Fig 3**. **Performance of different models trained on high-resolution manually annotated data.** Average training losses and Dice scores from four 3D segmentation models: VNet, UNETR, SwinUNETR, and nnUNet, evaluated using 5-fold cross-validation. A: Average Dice scores on the training, validation, and test sets for each model, with error bars indicating inter-fold variance; B: Training loss curves showing the average losses on the training and validation sets over epochs; C: Average validation Dice scores, with exponential moving average applied to each fold, for each model during training.

Fig 3B presents the average losses on the training (solid lines) and validation (dotted lines) datasets of each model during 5-fold cross-validation training. Fig 3C shows the average validation Dice score of five folds for each model, with a moving average applied as shown in the Eq 2, aligned with nnUNet [15].

$$val\_Dice_{ema} = 0.9 \times val\_Dice_{ema\_old} + 0.1 \times val\_Dice_{new}, \tag{2}$$

where the $val\_dice_{ema}$, $val\_Dice_{ema\_old}$ are the validation Dice of the current epoch and the previous epoch after exponential moving average, respectively. $val\_Dice_{new}$ denotes the validation Dice from the model of the current epoch. For the first epoch, the $val\_dice_{ema}$ is set to $val\_Dice_{new}$.

As shown in Fig 3B, average training losses across 5-fold of four models decreased steadily over 150 epochs, reaching 0.187 for VNet, 0.111 for UNETR, 0.099 for SwinUNETR, and 0.168 for nnUNet. The corresponding average validation losses were 0.182, 0.726, 0.211, and 0.186, respectively. The difference between average training and validation losses remained relatively small for VNet and nnUNet (within $\pm 0.05$) and moderate for SwinUNETR (within $\pm 0.1$), but was larger for UNETR ($\pm 0.6$). Notably, the UNETR average validation loss increased by 11.2% after epoch 90 to epoch 150 from 0.653 to 0.726. Fig 3C shows validation Dice scores after an exponential moving average applied during the training process. The nnUNet achieved the highest validation Dice score, followed by VNet, SwinUNETR, and UNETR. However, a decline in the validation Dice score can be seen on the UNETR after 90 epochs. The increase of a validation Dice score with a continuously declining training Dice score, coupled with a decrease of the exponentially moving average validation Dice [15], indicated that the UNETR experienced the overfitting problem.

Fig 4 visualises several validation samples from fold 0 with their ground truths and predictions from the four different methods. From these images, we saw that VNet, SwinUNETR and nnUNet can predict smooth glomeruli boundaries similar to the annotations, but UNETR had difficulty learning the correct boundary features. This negatively impacted the UNETR performance and resulted in low Dice scores. nnUNet outperformed other methods because fewer false positives were predicted, compared to other methods with plenty of false positive predictions, such as the first sample for UNETR, the second sample for SwinUNETR, and the fourth sample for VNet.

## Post-processing effects

We also proposed an automated post-processing pipeline to remove the outliers in the predictions and obtain clean pseudo-labels for the lower-resolution data. The nnUNet with the best performance on the annotated high-resolution cubes was selected as a benchmark method for correlative glomeruli segmentation, but its predictions when performing inference on the whole VOI of HiP-CT data can be clearly seen to have a large number of false positives (see Fig E in S1 Text), primarily falling into three areas within the kidney: fat, blood clots, and tubular structures, as shown in Fig 5.

During post-processing, parameter thresholds listed in Table 2 were sampled using LHS to eliminate outliers. To assess the performance of the sampled thresholds, we computed the Dice scores across all training cubes as well as in specific glomeruli-free cubes for both high and intermediate-resolution data. The parameter search results are presented in Fig F and Fig G in S1 Text. However, for the lowest-resolution data of complete organs, only the training cubes were utilised due to the larger VOIs with enough content (see Fig H in S1 Text).

A consistent minimum size threshold of 62 $\mu m$ in radius was applied across all resolutions. For data with high contrast between the kidney cortex and other structures against the background, thresholds for intensity variance and roundness were employed (at high and mid resolutions). In contrast, for the data with lower contrast (mid- and low-resolution overviews), neighbourhood density was used due to the sparse distribution of false positives compared to true positives. Additionally, a cortex mask (manually annotated) was applied to whole-kidney predictions to eliminate false positives occurring in the medulla and hilum regions, where glomeruli are absent. After post-processing, Dice scores were highly improved for high and intermediate-resolution data, while slightly improved for the low-resolution complete organ data.

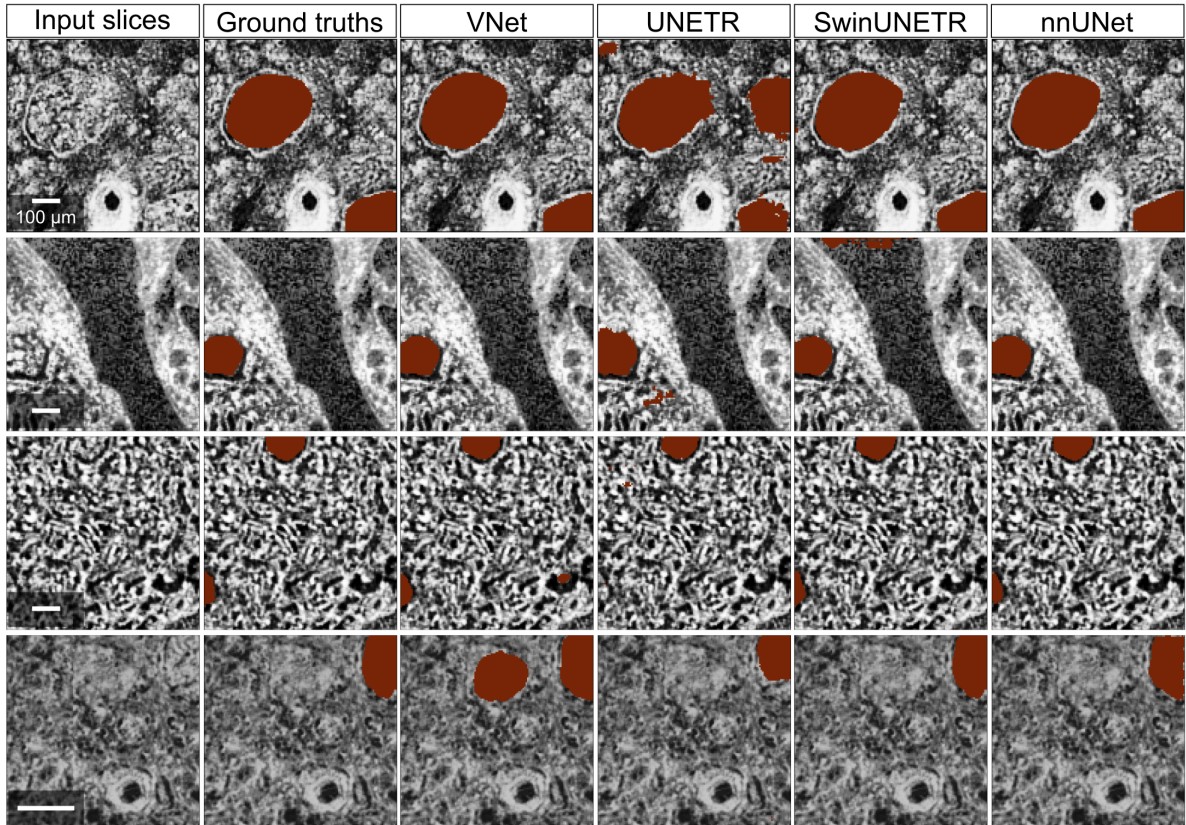

**Fig 4**. **Ground truths from manual annotations and predictions (both in red) from each model in fold 0 on high-resolution data, shown in 2D slices.** The 2D slices were sampled from validation cubes of each kidney; from top row down: 5 $\mu m$ S-20-28, 5.2 $\mu m$ LADAF-2021-17 left kidney, 5.2 $\mu m$ LADAF-2021-17 right kidney, and 2.58 $\mu m$ LADAF-2020-27 left kidney. The slices are of size $128^2$ and pre-processed by CLAHE.

At low-resolution data, the limited image quality restricted the differentiation between predicted glomeruli and the false positives in the cortical region. Although the proposed post-processing, consisting of size filters, neighbourhood density, and cortex mask, provided a feasible way to reduce the false positives, low contrast in the cortical area limited the applications of intensity-based and shape-based parameters, which are used in high and intermediate resolutions.

## Glomeruli segmentation on lower-resolution data

After training on high-resolution data with manual annotations and post-processing, nnUNet performed best and was used for correlative lower-resolution data. This section reports the results of LADAF-2020-27 left kidney at 12.1 $\mu m$/voxel and 25.08 $\mu m$/voxel.

To fine-tune the models, the predictions were registered, and the corresponding VOI in the lower-resolution data was cropped as pseudo-labelled training data. The correlative fine-tuning followed 5-fold cross-validation and the same patch size of $128^3$. As the glomeruli were small and sparsely distributed in the image, the training data could include many cubes without glomeruli. Those empty cubes could bias the model training process. Therefore, at the intermediate resolution(12.1 $\mu m$/voxel) where the glomeruli can still be clearly identified, we only kept 1.05% of the empty cubes after cropping. However, for the data at 25.08 $\mu m$/voxel that has unclear glomeruli boundaries, we found that removal of the empty cubes was not enough to train a model with better performance. Accordingly, instead of keeping empty cubes, we

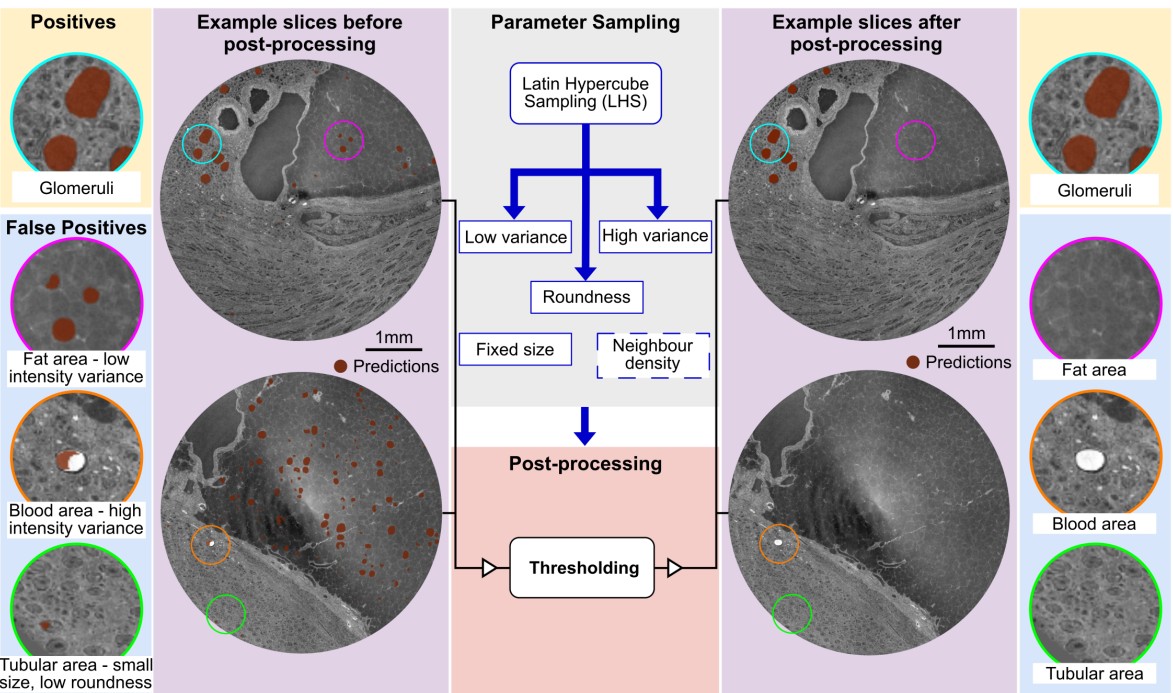

**Fig 5. Post-processing to reduce false positives in glomeruli segmentation.** False positives occurring in fat, blood clots, and tubular regions are removed using threshold-based filtering. Optimal thresholds were determined via Latin Hypercube Sampling (LHS). Representative examples are taken from the LADAF-2020-27 left kidney dataset.

**Table 2.** **Post-processing parameters and corresponding Dice scores (before and after post-processing) for predictions on LADAF-2020-27 left kidney scans at different resolutions.**

| Parameters | 2.58 $\mu m$/voxel prediction | 12.1 $\mu m$/voxel prediction | 25.08 $\mu m$/voxel prediction |
|---|---|---|---|
| Variance lower bound | 114770.713 | 179663.370 | N/A |
| Variance upper bound | 3069051.733 | N/A | N/A |
| Roundness | 0.709 | 0.682 | N/A |
| Size (radius in $\mu m$) | 62 | 62 | 62 |
| Density | N/A | W[1]=128, D[2]=3 | W=64, D=3 |
| Mask | N/A | N/A | Cortex mask applied |
| Dice scores (before/after) | 0.734 / 0.934 | 0.475 / 0.974 | 0.786 / 0.788 |

[1] windows size
[2] density of neighbourhood.

kept 0.7% of the cubes with label volume densities lower than 1%. The average number of patches used in fine-tuning is shown in Table 3.

The model was fine-tuned with 1000 epochs on the 12.1 $\mu m$/voxel data and 1500 epochs on the 25.08 $\mu m$ data, to reach the training plateau, as the training losses shown in Fig 6. For fine-tuning on the 25.08 $\mu m$/voxel data, an ablation study for the learning rate was performed, and it was found that by continuing to train 500 more epochs after 1000 epochs with a starting learning rate of 0.00074, performance was improved (see Fig D in S1 Text, for details on the training explorations). As shown in Table 3, the average Dice scores were calculated by applying the final training models of each fold on training and validation data splits. The model fine-tuned on the 12.1 $\mu m$/voxel data achieves an average Dice score of 0.958 on the training dataset, and 0.949 on the validation dataset. The model fine-tuned on the 25.08 $\mu m$/voxel

**Table 3**. **Training samples and average Dice scores of correlative 5-fold cross-validation training on lower-resolution data of LADAF-2020-27 left kidney with pseudo-labels.**

| Resolutions | 12.1 $\mu m$ | | 25.8 $\mu m$ | |
|---|---|---|---|---|
| | Train | Val | Train | Val |
| Average number of patches | 204 | 51 | 1054 | 263 |
| Average of Dice scores | 0.958 | 0.949 | 0.815 | 0.796 |
| Standard deviation of Dice scores | 0.003 | 0.006 | 0.002 | 0.002 |

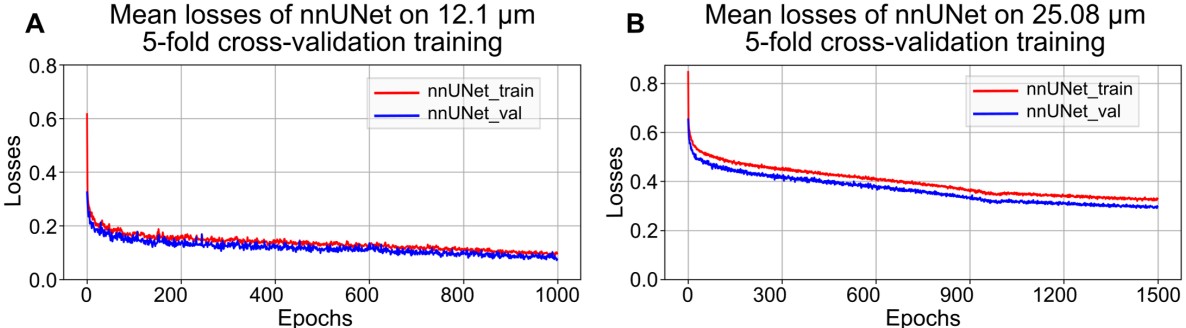

**Fig 6**. **Correlative training losses on lower resolution data of LADAF-2020-27 left kidney with pseudo-labels.** A: panel shows the mean losses of 5-fold cross-validation training at the intermediate resolution of 12.1 $\mu$m/voxel. B: panel presents the losses for the lowest resolution data of 25.08 $\mu$m/voxel.

data achieves scores of 0.815 and 0.796, respectively. To evaluate the fine-tuning effects from higher-resolution to lower-resolution VOIs, we conducted further experiments of comparison between fine-tuning the model and training a model from scratch. The results are displayed in Fig D in S1 Text, showing that fine-tuning improves the convergence rate and segmentation performance.

## Morphological analysis

Our pipeline on HiP-CT enables the morphological analysis of glomeruli across the whole human kidney. Having trained the model on high-resolution data, we applied the pipeline to two human kidneys: LADAF-2020-27 left kidney from a female donor aged 94, and LADAF-2021-17 right kidney [53] from a male donor aged 64. The training details of LADAF-2021-17 right kidney are presented in Table B and Fig I in S1 Text. The female donor suffered from hypertensive heart disease [54], microcrystalline arthritis (gout) [55], atrial fibrillation [56], and right cerebellar stroke [57], all of which can directly impact glomerular functionality. No kidney-related diseases were observed in the male donor. These two kidneys could provide comparable analysis across different ages, sexes, and healthy statuses.

As shown in Table 4, the total and cortical volumes of the LADAF-2020-27 left kidney were approximately 70 $cm^3$ and 40.6 $cm^3$, respectively, while those of the LADAF-2021-17 right kidney were approximately 136 $cm^3$ and 88.6 $cm^3$. These results indicated a notable difference in organ size, potentially attributable to donor age, sex, and health status

**Table 4**. **Kidney details for morphological analysis.**

| Kidneys | Sex | Age | Kidney volume ($cm^3$) | Cortical Volume ($cm^3$) | Number of glomeruli |
|---|---|---|---|---|---|
| LADAF-2020-27 Left | F | 94 | 69.74 | 40.56 | 231,179 |
| LADAF-2021-17 Right | M | 62 | 136.05 | 88.64 | 1,019,890 |

[58]. Additionally, the LADAF-2021-17 right kidney contained approximately 4.4 times more glomeruli (1,019,890 versus 231,179), further highlighting the substantial morphological disparities between the two donor kidneys.

Glomeruli predictions across the entire 3D human kidneys were presented in Fig 7A.1 and B.1, which are crucial for anatomical evaluation, as their spatial locations and volumes are linked to functional heterogeneity in filtration and susceptibility to disease [59]. By convolving a summation kernel of size $31^3$ on the glomeruli prediction centres, we generated the glomerular density distributions as shown in Fig 7A.2 and B.2 within the kidney cortex. The LADAF-2021-17 right kidney exhibited a higher maximum density for glomerular centres ($21.5/mm^3$, mean $9.4\pm4.9/mm^3$) compared to the LADAF-2020-27 left kidney ($16/mm^3$, mean $5.6 \pm 3.5/mm^3$), indicating denser glomerular distribution in the younger, healthy male kidney.

To further investigate the 3D spatial glomerular organisation within human kidneys, the cortical region was subdivided into three zones according to the glomerular diameters as in a previous study of [60]. The inner zone, located within three glomerular diameters from the medulla, corresponds to the juxtamedullary region, while the outer region, positioned within three glomerular diameters from the capsule, represents the superficial region [60]. The rest of the cortical area was classified as the middle zone (also known as the mid-cortical region). The tripartite division reflected the classical anatomical subdivision of the renal cortex into juxtamedullary, mid-cortical, and superficial layers, as described in [60,61]. Using the mean glomerular diameters from each kidney (205.14 $\mu m$ for LADAF-2020-27 left kidney and 182.68 $\mu m$ for LADAF-2021-17 right kidney), the cortical areas were patitioned into inner, middle, and outer zones, resulting in proportions of 11%, 75% and 14% for the LADAF-2020-27 left kidney and 10%, 79% and 11% for the LADAF-2021-17 right kidney, respectively, as presented in Fig 7C.1.

Across these three zones, most glomeruli were located in the middle zone as shown in Fig 7C.2, accounting for 91.34% in LADAF-2020-27 left kidney ($0.211 \times10^6$) and 86.8% in LADAF-2021-17 right kidney ($0.887 \times10^6$). In the LADAF-2021-17 right kidney, the inner and outer zones contained comparable numbers of glomeruli ($0.064 \times10^6$ vs. $0.07 \times10^6$, respectively). In contrast, the ageing and diseased LADAF-2020-27 left kidney showed an 88.9% reduction in the outer zone ($0.002 \times10^6$) relative to the inner zone ($0.018 \times10^6$).

Volumetric statistical analysis in Fig 7C.3 revealed distinct differences. For LADAF-2020-27 left kidney, the largest glomerular volumes occurred in the middle zone (median $0.431 \times10^7\mu m^3$; mean $0.455 \times10^7\mu m^3 \pm 0.218 \times 10^7\mu m^3$), followed by the inner zone (median $0.399 \times10^7\mu m^3$; mean $0.428 \times10^7\mu m^3 \pm 0.195 \times 10^7\mu m^3$) and the outer zone (median $0.382 \times10^7$; mean $0.407 \times 10^7\mu m^3 \pm 0.192 \times 10^7\mu m^3$). However, for the LADAF-2021-17 right kidney, inner and middle zones presented comparable volumes, with both medians $\sim 0.3 \times10^7\mu m^3$and means $\sim 0.32 \times10^7\mu m^3$. The glomerular size was noticeably smaller in the outer zone (median $0.281 \times10^7\mu m^3$; mean $0.295 \times10^7\mu m^3 \pm 0.102 \times 10^7\mu m^3$). By comparing median glomerular volumes across the kidneys, the results revealed that the glomerular volumes increased 30.4%, 42.7% and 35.9% in the inner, middle and outer zones from LADAF-2020-27 left to LADAF-2021-17 right, respectively, indicating substantial enlargement of glomeruli in the ageing and renal-diseased kidney. Because glomerular volume distributions deviated from normality, zonal differences were assessed using the Kruskal–Wallis test across the three cortical zones. The analysis showed a highly significant zonal difference in glomerular size ($p < 0.001$) on both kidneys.

Since the segmentation model achieved an average Dice score of approximately 0.8 on both kidneys at the complete kidney resolution, we extended the morphological analysis to the intermediate-resolution VOIs, where a 0.95 Dice score was reached. As shown in Fig 7D.1, the zonal proportions of the intermediate-resolution VOIs remained consistent with the middle zone still dominant (82% LADAF-2020-27; 80% for LADAF-2021-17). However, the outer zone proportion decreased in LADAF-2020-27 to 7% and increased slightly in LADAF-2021-17 to 11%. Glomerular amounts, as shown in Fig 7D.2, presented similar trends, with the middle zones predominant. By comparing the medians and means of glomerular volumes in intermediate-resolution VOIs, similar patterns, like complete organs, were also presented from Fig 7D.3. On the one hand, LADAF-2020-27 left kidney observed larger glomeruli in the middle zone, while the glomerular sizes in the inner and outer zones were close. As for LADAF-2021-17 Right kidney, middle and inner zones were comparable, while a

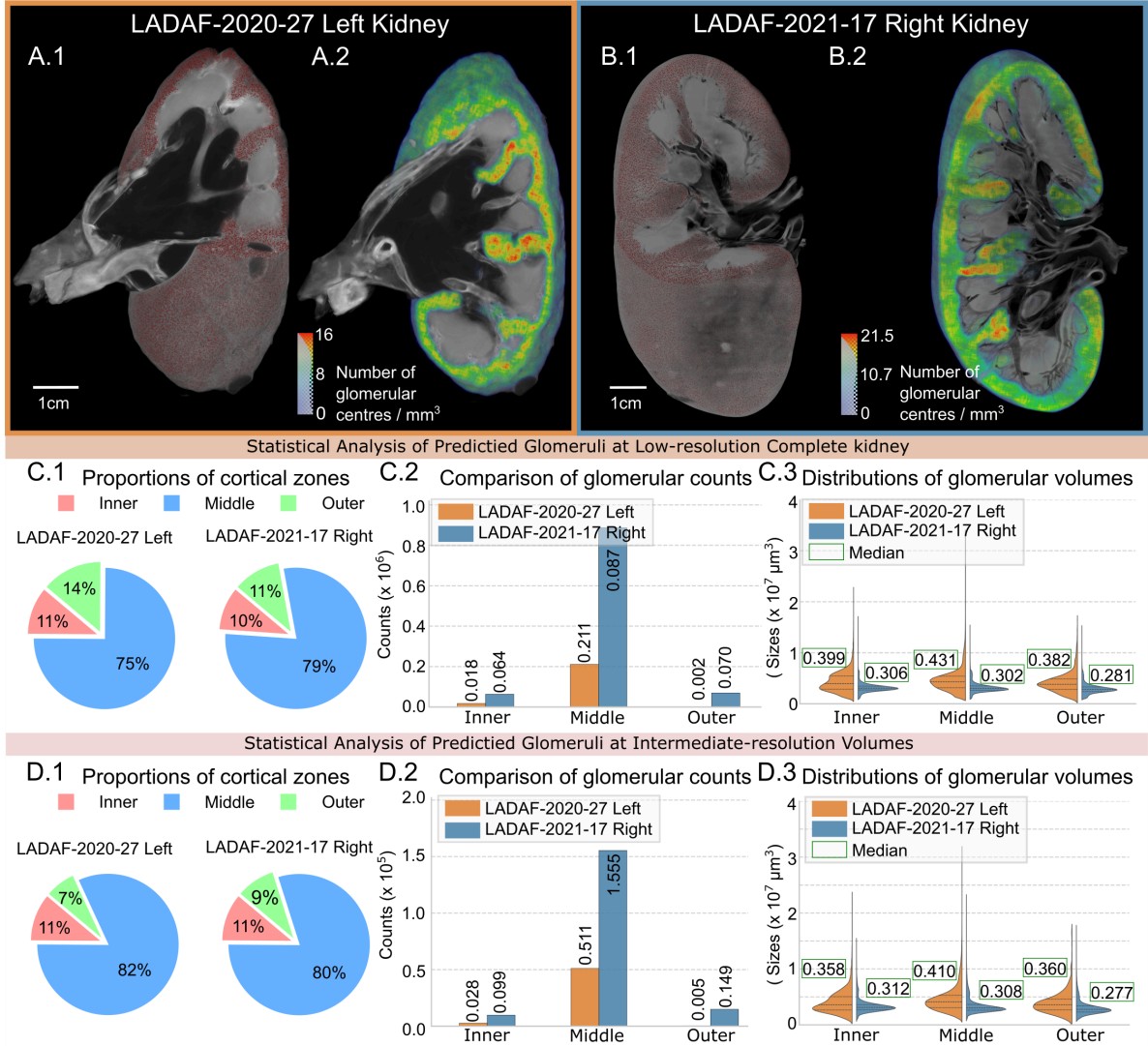

**Fig 7**. **Glomeruli morphological analysis on the complete kidneys and intermediate-resolution VOIs.** (A.1) and (A.2) present 3D renderings of predicted glomeruli and the glomerular centre density calculated by a summation kernel on the HiP-CT of the complete kidney (LADAF-2020-27 Left, Female, aged 94). Similarly, (B.1) and (B.2) are for LADAF-2021-17 Right kidney (Male, aged 62). (C.1) to (C.3) depicts the glomerular statistical analysis on complete low-resolution kidneys, including the proportions of different cortical zones (C.1) defined by glomerular diameters, the number of glomeruli in each zone (C.2), and volumetric distributions across different zones (C.3). (D.1) to (D.3) present statistical analysis on intermediate-resolution VOIs, which are identical to complete organs.

decrease can be observed in the outer zones. On the other hand, the glomerular volume was greater in the LADAF-2020-27 left kidney, similar to the results from complete organ analysis. Overall, the morphological analysis on intermediate-resolution VOIs, which achieved a higher Dice score, validated the distribution patterns from the low-resolution complete kidney, but provided more accurate size estimation. The results reinforced the robustness of our multiscale segmentation approach on HiP-CT data. The Kruskal–Wallis test across the three cortical zones also confirmed a highly significant zonal effect on glomerular size ($p < 0.001$) on both intermediate-resolution VOIs.

## Discussion

### Computational constraints and resolution effects

HiP-CT is an inherently isotropic 3D modality, creating individual images that range in size from gigabytes to terabytes. As a result, training deep learning models on this data demands significant computational resources. The challenge is balancing the richness of contextual information in the inputs with the constraints of computational memory. Therefore, the training patches dimension was set to $128^3$ (filling 2GB of memory), which is large enough to contain complete glomeruli in high-resolution data while fitting into available memory for a batch size of two. The model performs best on the intermediate-resolution data, achieving a mean validation Dice score of 0.949, as more complete glomeruli with relatively clear boundaries are involved and more contextual information surrounding the glomeruli was provided in this resolution, compared to the other two resolutions, which have mean validation Dice scores of 0.860 and 0.796.

### HiP-CT contrast enhancement

Contrast enhancement as a pre-processing step for HiP-CT data plays a key role in training a successful segmentation model. As shown by a recent Kaggle competition to segment kidney vasculature on HiP-CT data [6], the ranges of voxel intensities are narrow and change between samples and resolutions due to factors including the X-ray scan setup, anatomical differences, and sample preparation, which are different for each sample and resolution. This results in the shift and bias of the model weights if the training dataset involves several samples. Therefore, we applied CLAHE, which can enhance local contrast while maintaining overall brightness, making small structures, such as glomeruli, more distinguishable. However, we used default hyperparameters when applying CLAHE to different samples. Although this helps generate a similar and wide range of voxel intensity for each sample, future works could explore the statistical impact of varying these parameters or investigate alternative contrast enhancement techniques applied to HiP-CT data. This will be particularly important for extending the model to larger cohorts of HiP-CT kidney data collected over recent years, during which the HiP-CT imaging methodology has evolved, and data quality has significantly improved.

### Model overfitting problem

In the first hierarchical segmentation cycle, we benchmarked four different deep learning models. From the experiment results, UNETR was found to experience an overfitting problem in this task. On the one hand, the annotations of glomeruli were sparse, where the mean proportion of voxels annotated as glomeruli within a $128^3$ training cube was 3.5% $\pm$ 3.0% for the high-resolution data. In such a sparse dataset, a complicated transformer-based model with a large number of trainable parameters, such as UNETR (121.4M), can be difficult to optimise [62]. By contrast, nnUNet (31.2M), SwinUNETR (15.7M) and VNet (9.3M) tend to be easier to train. On the other hand, due to the glomeruli being small, insufficient long-range dependencies can also cause the overfitting problem for a complicated transformer-based UNETR.

### Error accumulation and improvements

The proposed multiscale segmentation pipeline, whether applied to glomeruli as in this study or applied to other organ structures as a more general concept, can lead to cumulative errors introduced during the generation of pseudo-labels from higher-resolution predictions. The errors arise from several factors, such as segmentation inaccuracies in the model at previous scales, misalignments during multi-resolution registration, and the loss of structural details due to decreased resolution from the VOI scans up to the complete organ scans. As the pipeline progresses across scales, these errors can accumulate and become more noticeable, ultimately reducing the reliability of the segmentations at the lowest resolution. To address this, we developed an automated post-processing technique to eliminate false positives based on multiple intensity-based and morphological properties, searched by LHS according to different scales, and validated against (pseudo-)ground truths.

In addition to post-processing, another key strategy for reducing error accumulation in the multiscale segmentation pipeline is improving imaging quality. All data used in this study were acquired on the beamline BM05 in the period from 2020 - 2022, before the start of the new BM18 beamline. The increased capabilities of beamline BM18, notably the increase in propagation distance for lower-resolution scans and the larger beam size, mean that the speed and contrast sensitivity of scans can be higher than those used to develop this model [63]. For example, a whole kidney can be overviewed at 10 $\mu m$/voxel on BM18, or a 20 $\mu m$/voxel scan can be performed with 20 meters of propagation (as opposed to 3.5 meters), significantly enhancing the contrast sensitivity and thus clarity of the images. Using these scanning conditions on beamline BM18 enables higher throughput experiments, allowing multiple kidneys to be imaged in a single acquisition under more consistent image conditions. This reduces the sample-to-sample variability, simplifies the pre-processing requirements and improves the robustness of the segmentation model. An intact organ in a higher voxel size and better resolution helps reduce the number of resolution transitions within the multiscale segmentation pipeline. Indeed, we have already collected such imaging data and are actively applying the pipeline developed in this study to a larger number of studies to investigate general trends in glomerular morphology and spatial distribution.

Whilst higher imaging quality and consistency are always fundamental to the biological conclusions and applications for this multiscale segmentation, the challenge of feature degradation is also inherent to this approach. As resolution decreases across the hierarchical pipeline, image feature degradation limits the visibility of the fine features (as illustrated in Fig 1). In our case, this degradation affects glomeruli, but as the quality of images increases, the segmented target at the highest resolution might change to smaller structures, e.g. single cells. Despite better imaging quality, propagating segmentations of such fine structures across scales will face the same challenge. Therefore, segmentation accuracy can be compromised due to pixelation and the limitation of contrast sensitivity with lower-resolution data.

To mitigate these challenges, advanced feature enhancement, such as a super-resolved tomographic reconstruction technique [64], has been proposed. However, it is important to acknowledge that this multiscale segmentation approach cannot be infinitely propagated to lower-resolution data simply through image enhancement. At some point, the lower-resolution data lacks sufficient statistical information to support segmentations of specific features. This limitation is closely tied to the constraints of generative models and the challenge of hallucination [65]. Fundamentally, the model presented in this study serves not only as a practical multiscale segmentation tool but also as a framework to evaluate when the feature can be confidently inferred from lower-resolution data through image enhancement and when higher-resolution imaging becomes necessary.

## Glomerular morphological analysis

The human kidney is traditionally thought to contain approximately one million glomeruli [66], though more recent studies have revealed substantial variability, with up to a 13-fold difference reported among healthy individuals [67]. Autopsy studies indicate that nephron number tends to be lower in hypertensive individuals compared to age-matched normotensive controls [68]. Age-related nephron loss is also well-documented, with an estimated decline of ca. 3,676 glomeruli per kidney per year after age 18 [69]. In hypertensive patients, this loss may accelerate to ca. 200,000 glomeruli per decade after age 50, emphasising the combined impact of ageing and hypertension on nephron number [69].

The average glomerular volumes observed in our healthy male kidney (LADAF-2021-17 right kidney) are approximately 0.32 $\times 10^7 \mu m$ in inner and middle zones, while 0.295 $\times 10^7 \mu m$ in the outer zone. This result aligns well with values reported by Denic et al. [61], which performed analysis on kidney biopsy histology showing 0.29 $\times$ 10$^7$ $\pm$ 0.11 $\times$ 10$^7$ $\mu m^3$, 0.32 $\times 10^7 \pm 0.12 \times 10^7$ $\mu m^3$, and 0.25 $\times 10^7 \pm 0.11 \times 10^7$ $\mu m^3$ in juxtamedullary, mid-cortical, and superficial regions, respectively. Hughson et al. [70] reported that hypertension typically leads to a 20–30% increase in glomerular volume. Our findings are consistent with this trend, but show slightly higher increases of 35.4%, 41.7% and 38% in the inner, middle and outer zones, respectively. These greater enlargements likely reflect additional contributing factors, such as age, beyond

hypertension individually. In the hypertensive female donor, glomerular size was elevated across all cortical regions, suggesting that hypertension induces global glomerular hypertrophy rather than region-specific effects.

## Conclusion

Studies on hierarchical biomedical systems are critical to discovering fundamental physiological functions across multiple scales. This requires advanced imaging techniques to produce images across complete macro-scale organs and micro-scale near-cellular structures, and effective multiscale image processing pipelines to enable integrated analysis across these resolutions. In this work, we used HiP-CT, which can image intact human organs at ca. 20 $\mu m$/voxel down to high-resolution VOIs at ca. 1 $\mu m$/voxel. It inherently provides multiscale biomedical image datasets that capture the tiny functional structures at different resolutions. Using glomeruli segmentation in the human kidney as a case study, we developed a hierarchical segmentation pipeline based on deep neural networks. Our results demonstrated that the segmentations can be propagated from high-resolution VOIs to entire low-resolution kidneys, enabling anatomical morphology analyses of those tiny functional units like glomeruli. While this work focused on human kidneys, the principles and architecture are generalisable to other organs that also exhibit similar multiscale structures involving small functional units across entire organs.

## Supporting information

**S1 Text. Supplementary Text, Figures and Tables.**
(PDF)

## Acknowledgments

The authors would like to acknowledge ESRF beamtimes md1252, md1290, and md1389 as sources of the data, as well as EPSRC grant JADE-2 [EP/T022205/1] for HPC resources.

## Author contributions

**Conceptualization:** Paul Tafforeau, Peter D. Lee, Claire L. Walsh.

**Data curation:** David Stansby, Saskia Carroll.

**Funding acquisition:** Peter D. Lee, Claire L. Walsh.

**Investigation:** Yang Zhou, Shahab Aslani, Yousef Javanmardi, Joseph Brunet, Alexandre Bellier, Maximilian Ackermann, Paul Tafforeau.

**Methodology:** Yang Zhou, Joseph Brunet, Paul Tafforeau, Claire L. Walsh.

**Project administration:** Peter D. Lee, Claire L. Walsh.

**Resources:** Alexandre Bellier.

**Supervision:** Peter D. Lee, Claire L. Walsh.

**Validation:** Yang Zhou.

**Visualization:** Yang Zhou.

**Writing – original draft:** Yang Zhou, Shahab Aslani, Yousef Javanmardi.

**Writing – review & editing:** Yang Zhou, Shahab Aslani, Yousef Javanmardi, Joseph Brunet, David Stansby, Saskia Carroll, Alexandre Bellier, Maximilian Ackermann, Paul Tafforeau, Peter D. Lee, Claire L. Walsh.

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
