## [Decision Letter · Decision Letter 0]

25 Aug 2025

PCOMPBIOL-D-25-01047

Multiscale Segmentation using Hierarchical Phase-contrast Tomography and Deep Learning

PLOS Computational Biology

Dear Dr. Zhou,

Thank you for submitting your manuscript to PLOS Computational Biology. After careful consideration, we feel that it has merit but does not fully meet PLOS Computational Biology's publication criteria as it currently stands. Therefore, we invite you to submit a revised version of the manuscript that addresses the points raised during the review process.

Please submit your revised manuscript within 60 days Oct 25 2025 11:59PM. If you will need more time than this to complete your revisions, please reply to this message or contact the journal office at ploscompbiol@plos.org. Please include the following items when submitting your revised manuscript:

We look forward to receiving your revised manuscript.

Kind regards,

Jinshan Xu

Academic Editor

PLOS Computational Biology

Pedro Mendes

Section Editor

PLOS Computational Biology

**Journal Requirements:**

2) Please ensure that your ethics statement is included under a subheading 'Ethics Statement', at the beginning of your Methods section.

Note: The Ethics Statement should include : The full name(s) of the Institutional Review Board(s) or Ethics Committee(s), the approval number(s), or a statement that approval was granted by the named board(s), and a statement that formal consent was obtained (must state whether verbal/written).

Potential Copyright Issues:

i) Figure 2. Please confirm whether you drew the images / clip-art within the figure panels by hand. If you did not draw the images, please provide (a) a link to the source of the images or icons and their license / terms of use; or (b) written permission from the copyright holder to publish the images or icons under our CC BY 4.0 license. Alternatively, you may replace the images with open source alternatives. See these open source resources you may use to replace images / clip-art:

1) State the initials of the author (s) who received this grant 2022-316777. For example: "This work was supported by the National Institutes of Health (####### to AM; ###### to CJ) and the National Science Foundation (###### to AM)."

3) If any authors received a salary from any of your funders, please state which authors and which funders.

6) Thank you for stating "No competing interests among authors." Please modify your Competing Interest statement on the submission form to the standard "The authors have declared that no competing interests exist."

7) Please label the supplementary tables and figures as “S1 Table” and “S2 Table,” "S1 Figure", S2 Figure" and so forth.

**Reviewers' comments:**

Reviewer's Responses to Questions

**Comments to the Authors:**

**Please note that one of the reviews is uploaded as an attachment.**

Reviewer #1: uploaded as attachment

Reviewer #2: This paper proposes a novel multi-scale segmentation pipeline that enables the segmentation of micro-scale functional units across entire organs, thereby facilitating the analysis of functional relationships and spatial distributions within the context of whole-organ anatomy. The proposed framework is innovative, particularly in its use of generated histograms of morphological and spatial properties to filter outliers. The authors conducted extensive experiments and have made both their dataset and code publicly available. This work contributes to the enhanced utilization of HiP-CT and demonstrates significant potential for medical applications. The reviewer encourages the authors to address the following concerns:

1. The necessity and effectiveness of fine-tuning and transfer learning are not clear. In terms of feature extraction, the feature space differs substantially across resolutions. The reviewer recommends that the authors provide additional validation regarding the effectiveness of fine-tuning a model trained on high-resolution data when applied to low-resolution data.

2. The overfitting behavior observed in UNETR lacks a thorough analysis. As shown in Figure 3, UNETR exhibits more pronounced overfitting compared to the other three methods. The reviewer suggests that the authors include an analysis of the possible causes of this phenomenon.

Reviewer #3: The authors describe a multiscale strategy for deep learning with high, and low resolution images, here HiP-CT scans of the kidney.

Their strategy is to mutiscale across scales in the following way: High-resolution predictions are propagated to lower resolutions, which creates training pseudo-labels for models specific to coarser scales. In this way, they are able to achieve segmentation not only at organ-level but also a glomerular levels.

This is a promising approach that can move the field forward.

My difficulty in reading the paper was they introduce the various components of their pipeline woven into the numerous sections in the paper. There is no clear outline of the proposed strategy and what networks they use in each stage. Further, the terminology "pseudo-labels" was very confusing as the authors never quite define what they mean.

Overall, the paper has a lot of valuable details to offer for the interested practitioner, but it is difficult to read this without first having some bearings as to what the authors wanted to achieve and how did they plan to go about it.

**Have the authors made all data and (if applicable) computational code underlying the findings in their manuscript fully available?**

Reviewer #1: Yes

Reviewer #2: Yes

Reviewer #3: Yes

PLOS authors have the option to publish the peer review history of their article (what does this mean?). If published, this will include your full peer review and any attached files.

Reviewer #1: **Yes:** Jens J. G. Lohmann

Reviewer #2: **Yes:** Tao Chen

Reviewer #3: No

**Figure resubmission:**
---

## [Decision Letter · Decision Letter 1]

16 Jan 2026

Dear Dr Zhou,

We are pleased to inform you that your manuscript 'Multiscale Segmentation using Hierarchical Phase-contrast Tomography and Deep Learning' has been provisionally accepted for publication in PLOS Computational Biology.

Best regards,

Jinshan Xu

Academic Editor

PLOS Computational Biology

Pedro Mendes

Section Editor

PLOS Computational Biology

Reviewer's Responses to Questions

**Comments to the Authors:**

Reviewer #1: Thanks to the authors for addressing my comments in such detail!

I have no further comments to add.

**Have the authors made all data and (if applicable) computational code underlying the findings in their manuscript fully available?**

Reviewer #1: Yes

PLOS authors have the option to publish the peer review history of their article (what does this mean?). If published, this will include your full peer review and any attached files.

Reviewer #1: **Yes:** Jens J G Lohmann

---

## [Editor Report · Acceptance letter]

PCOMPBIOL-D-25-01047R1

Multiscale Segmentation using Hierarchical Phase-contrast Tomography and Deep Learning

Dear Dr Zhou,

I am pleased to inform you that your manuscript has been formally accepted for publication in PLOS Computational Biology. Your manuscript is now with our production department and you will be notified of the publication date in due course.

With kind regards,

Anita Estes
